# Long-term outcomes of survivors of neonatal insults: A systematic review and meta-analysis

Dorcas N. Magai[ID][1,2]\*, Eirini Karyotaki[ID][2], Agnes M. Mutua[ID][1], Esther Chongwo[1], Carophine Nasambu[1], Derrick Ssewanyana[ID][1,3], Charles R. Newton[1,4,5], Hans M. Koot[ID][2], Amina Abubakar[ID][1,4,5,6]

**1** Centre for Geographic Medicine Research Coast, Kenya Medical Research Institute, Kilifi, Kenya, **2** Department of Clinical, Neuro- and Developmental Psychology, Amsterdam Public Health Research institute, Vrije Universiteit Amsterdam, The Netherlands, **3** Utrecht Centre for Child and Adolescent Studies, Utrecht University, Utrecht, The Netherlands, **4** Department of Public Health, Pwani University, Kilifi, Kenya, **5** Department of Psychiatry, University of Oxford, Oxford, England, United Kingdom, **6** Institute for Human Development, Aga Khan University, Nairobi, Kenya

\* dmagai@kemri-wellcome.org

**Data Availability Statement:** The data underlying the results presented in the study are available from https://doi.org/10.7910/DVN/NPYQBL.

## Abstract

### Background

The Millennium Developmental Goals ensured a significant reduction in childhood mortality. However, this reduction simultaneously raised concerns about the long-term outcomes of survivors of early childhood insults. This systematic review focuses on the long-term neuro-cognitive and mental health outcomes of neonatal insults (NNI) survivors who are six years or older.

### Methods

Two independent reviewers conducted a comprehensive search for empirical literature by combining index and free terms from the inception of the databases until 10th October 2019. We also searched for additional relevant literature from grey literature and using reference tracking. Studies were included if they: were empirical studies conducted in humans; the study participants were followed at six years of age or longer; have an explicit diagnosis of NNI, and explicitly define the outcome and impairment. Medians and interquartile range (IQR) of the proportions of survivors of the different NNI with any impairment were calculated. A random-effect model was used to explore the estimates accounted for by each impairment domain.

### Results

Fifty-two studies with 94,978 participants who survived NNI were included in this systematic review. The overall prevalence of impairment in the survivors of NNI was 10.0% (95% CI 9.8–10.2). The highest prevalence of impairment was accounted for by congenital rubella (38.8%: 95% CI 18.8–60.9), congenital cytomegalovirus (23.6%: 95% CI 9.5–41.5), and hypoxic-ischemic encephalopathy (23.3%: 95% CI 14.7–33.1) while neonatal jaundice has the lowest proportion (8.6%: 95% CI 2.7–17.3). The most affected domain was the neurodevelopmental domain (16.6%: 95% CI 13.6–19.8). The frequency of impairment was highest

**Funding:** DNM was supported through the DELTAS Africa Initiative [DEL-15-003]. The DELTAS Africa Initiative is an independent funding scheme of the African Academy of Sciences (AAS)'s Alliance for Accelerating Excellence in Science in Africa (AESA) and supported by the New Partnership for Africa's Development Planning and Coordinating Agency (NEPAD Agency) with funding from the Wellcome Trust [107769/Z/10/Z] and the UK government. The views expressed in this publication are those of the authors and not necessarily those of AAS, NEPAD Agency, Wellcome Trust, or the UK government. No funding bodies had any role in the study design, data collection and analysis, decision to publish, or preparation of the manuscript.

**Competing interests:** The authors have declared that no competing interests exist.

for neurodevelopmental impairment [22.0% (*IQR* = 9.2–24.8)] and least for school problems [0.0% (*IQR* = 0.0–0.00)] in any of the conditions.

## Conclusion

The long-term impact of NNI is also experienced in survivors of NNI who are 6 years or older, with impairments mostly experienced in the neurodevelopmental domain. However, there are limited studies on long-term outcomes of NNI in sub-Saharan Africa despite the high burden of NNI in the region.

## Trial registration

Registration number: CRD42018082119.

## Introduction

Neonatal insults (NNI), defined as injury during the first 28 days of life, are associated with neonatal mortality, morbidity, and adverse neurodevelopmental outcome [1]. Globally, 5.9 million child deaths occur annually during the perinatal and neonatal periods [2, 3], accounting for 43% of deaths in children younger than 5 years of age [4]. Out of the total child-deaths that occur during this period, almost three-quarters occur in sub-Saharan Africa (SSA) and South Asia [5]. As part of the Millennium Developmental Goals, there was a global commitment to reduce childhood mortality by two-thirds between 1990 to 2015 [6]. Although the target was not fully achieved, there was a significant reduction in annual childhood mortality from 12.7 million in 1990 to 5.7 million in 2015 [7]. This reduction has increased attention about the long-term outcomes that may affect the functioning and quality of life of those children who survived early childhood conditions, and especially NNI (5).

Globally, the most common NNI include sepsis, congenital meningitis, hypoxic-ischemic encephalopathy (HIE), preterm birth, neonatal jaundice (NNJ), cytomegalovirus infection (CMV), herpes, congenital rubella, and toxoplasmosis [8]. Children who survive NNI are likely to develop adverse long-term outcomes such as neurocognitive impairment, developmental delay, hearing and visual impairment, cerebral palsy, and behavioral and emotional problems [9–15]. For instance, children who survived CMV were reported to develop sensorineural hearing loss and neurodevelopmental impairment [9], while congenital rubella is associated with hearing impairment [16, 17]. Intrauterine growth restriction (IUGR) is associated with neurodevelopmental impairment in the early years [12]; HIE was associated with motor and developmental delay during infancy [10], and NNJ was associated with adverse neurodevelopmental outcomes [8, 18–22].

There are few systematic reviews on the long-term outcomes of NNI. A review by Mwaniki et al. [8] included 153 studies with 22161 survivors of intrauterine insults with a follow-up period of at least 6 months and found that 39% of children who survive intrauterine or neonatal insults develop at least one long-term sequelae. The authors found that the survivors are likely to develop neurocognitive problems (cognitive impairment, developmental delays, and learning difficulties), visual and hearing impairment, and cerebral palsy. Mwaniki and colleagues also reported that congenital rubella and HIE had the highest prevalence of long-term sequelae (37% and 81% respectively) while NNJ has the lowest risk for long-term sequelae (18%).

Mwaniki et al.'s study reviewed studies with both short-term and long-term outcomes (from 6-months-old infants and older children); however, less is known about the long-term outcomes in school-aged children or older age groups globally who survived NNI [8]. As children grow older, their brains may compensate for brain injury during the neonatal period–a phenomenon termed brain plasticity [23]. Therefore NNI-associated impairments reported during early childhood may resolve as children grow older; it is, therefore, essential to identify on the residual neurodevelopmental sequelae at school-age and older age, which may persist and interfere with education, employment, and social functioning. This systematic review focuses on the long-term neurocognitive and mental health outcomes of NNI survivors who are six years or older.

## Methods

### Literature search

We conducted a comprehensive search for empirical literature in the following databases: PubMed, PsycINFO, Web of Science, Embase, and CINHAL. Relevant grey literature was searched in: ERIC, Open Grey, The Health Care Management Information Consortium (HMIC) database, the National Technical Information Service, and PsycEXTRA. We also used reference tracking to search for additional relevant literature. The detailed search terms (formulated by DNM and AA) comprised the index, and free terms of the different neonatal insults and the outcome measures combined with Boolean operators ("OR" and "AND") were used each database (S1 Appendix). The literature search included studies conducted from 1947 to 10th October 2019. This systematic review is registered in PROSPERO (https://www.crd.york.ac.uk/prospero/display_record.php?RecordID=82119); registration number CRD42018082119.

### Inclusion and exclusion criteria

Four reviewers (DNM, CN, EC, and AMM) independently screened the articles by titles, abstract, and full text for eligibility. Criteria for eligible studies were: i) empirical studies conducted in humans; ii) the study participants had follow-up data coinciding with the age bracket at 6 years or older (we considered children from six years as this is the age where most children join elementary school, especially in low-and-middle-income countries); iii) explicit diagnostic criteria for the neonatal insult, and iv) precise definition of the outcome and impairment (based on either a comparison to a control or a standardized test). Studies were excluded if i) it was not clear if the children had the diagnosis of NNI during the neonatal period (i.e., the first 28 days of life); ii) participants had another comorbid neural tube defect conditions, e.g. spina bifida; and iii) if the studies did not make a comparison to either a control or standardized test to define impairment in the outcomes.

### Data extraction and quality assessment

Information extracted from the studies included: the author's name and year of publication, the country where the study was conducted, the sample size, age of participants at follow-up, study design, the type of neonatal insult, the assessment tools used, the neurocognitive or mental health outcome assessed, and a general summary of the study findings (see S1 Table). Quality assessment of the studies in this review was guided by the Newcastle-Ottawa scale (NOS) for assessment of the quality of cohort studies [24]. This quality assessment tool assigns a maximum of four points for selection, two points for comparability, and three points for exposure or outcome. The ranking of the total scores is as follows: 0–3 for low-quality studies, 4–6 for

moderate quality, and 7–9 for high-quality studies. Both data extraction and quality assessment of the studies were independently done by two reviewers (DNM and AMM). In cases where the reviewers disagreed on articles included, disagreements were resolved by reassessment of the studies, discussions, and a third reviewer where necessary.

### Data analysis

Descriptive statistics such as frequencies and percentages were used to describe the outcomes in different impairment domains for each neonatal insult, the geographic distribution of studies, and the NNI reported. Medians and interquartile range (IQR) of the proportions of survivors of the different NNI with any impairment were calculated.

The outcome variable was categorized into 6 major domains of impairment: i) neurodevelopmental impairment (cognition, language, developmental delay, memory); ii) hearing impairment; iii) vision impairment; iv) neurological impairment (cerebral palsy, clinical, and motor impairment); v) epilepsy; and vi) mental/behavioral problems. The criteria for the outcomes assessed and instruments used are provided in S1 Table. The variation in effect sizes attributed to heterogeneity was explored using $I2$ statistic of the DerSimonian and Laird method [25]. The 95% confidence interval (95% CI) around the $I2$ was used to express the uncertainty associated with the $I2$. A random-effect model was then used—after Freeman-Turkey Double Arcsine transformation—to explore the estimates accounted for by each impairment domain. Impairment domains represented by less than 3 studies per condition were excluded from this analysis. Data were analyzed using STATA (version 15) [26].

To explore the confounding effects of follow-up periods (6–12 years, 13–18 years, and >18 years), study design (prospective vs retrospective), and region (North America, Asia, Europe, and Oceania) on the risk of impairment, a meta-regression was conducted for preterm birth only as other conditions had less than ten studies. To identify extreme findings, a visual inspection of the funnel plot was done, and Begg's adjusted rank correlation explored.

## Results

Fifty-two eligible studies (S1 Appendix) were included in this systematic review following a screening process that involved 80,038 articles (see Fig 1). Of the 52 eligible studies, 44.2% had an unexposed comparison group, while 55.8% only included an exposed group. For studies that had control groups, we report only on the case group (i.e. children who were diagnosed with NNI). More than half of the eligible studies were conducted in Europe (59.6%), 26.9% were conducted in North America, 7.7% were conducted in Asia, 5.8% were conducted in Oceania countries, and none in Africa. Most of the studies (76.9%) had a prospective study design, while 23.1% used a retrospective study design. A table with a summary of the characteristics of the studies is provided in S1 Table. The quality assessment of the 52 eligible studies indicates that 30 studies (57.7%) were categorized as "medium" quality while 22 studies (42.3%) were categorized as "high" quality (see S2 Table).

Overall, 94,978 survivors of NNI were included in the studies in this review. The median number of participants per study was 125 ($IQR$ = 78.5–281.5). The median age at follow-up was 8 years (6.0 to 10.0) years. The proportion of impairment among study participants aggregated by the form of NNI is summarized in Table 1. The most examined or studied NNI were HIE (15.4%), NNJ (15.4%), and preterm birth (42.3%). The overall prevalence of impairment in the survivors of NNI was 10.0% (95% CI 9.8–10.2). The highest prevalence of impairment was accounted for by congenital rubella (38.8%: 95% CI 18.8–60.9), CMV (23.6%: 95% CI 9.5–41.5), and HIE (23.3%: 95% CI 14.7–33.1) while NNJ has the lowest proportion (8.6%: 95% CI 2.7–17.3). The most affected domain was the neurodevelopmental domain (16.6%: 95% CI

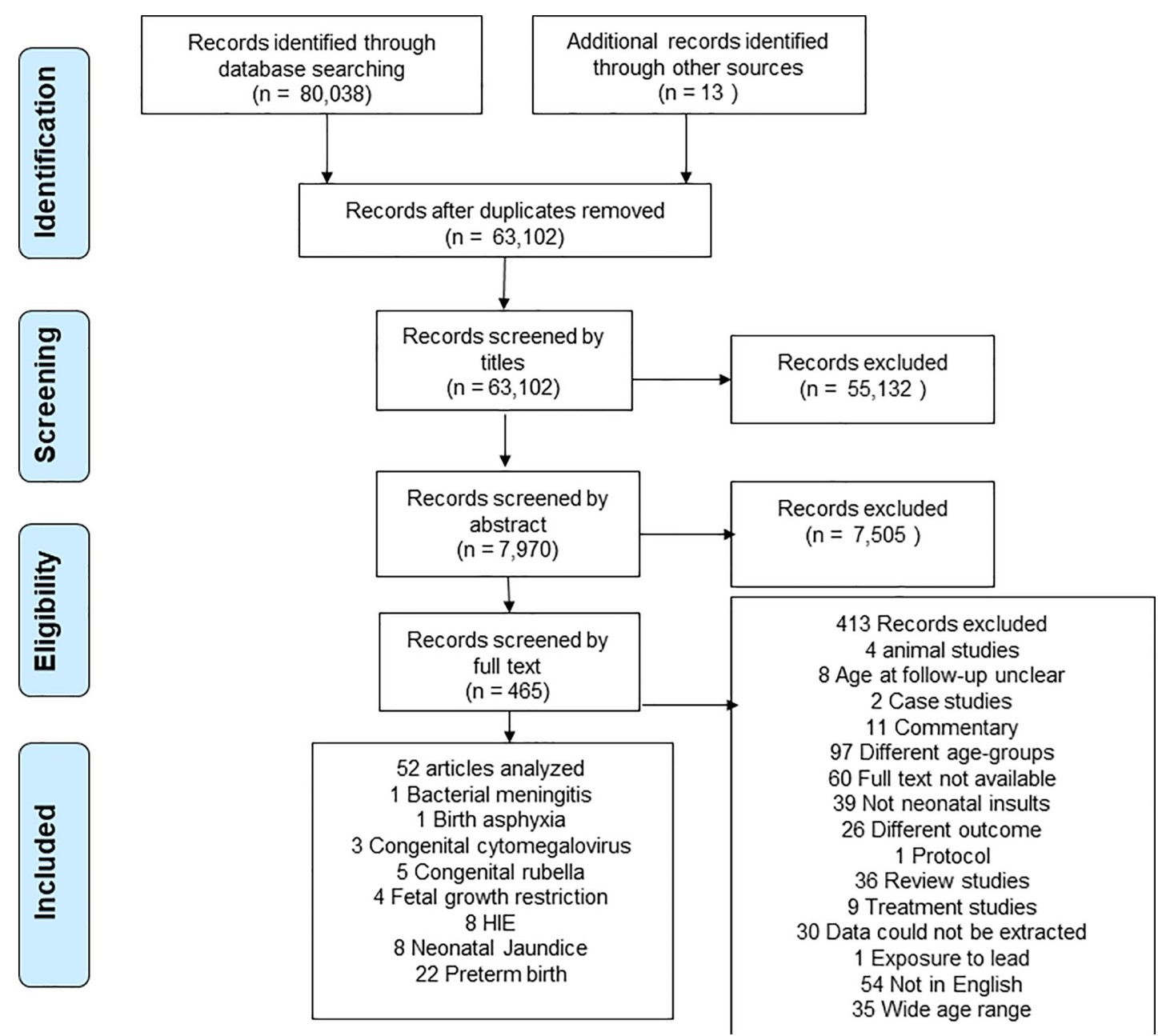

**Fig 1. Flow chart showing the selection of studies on long-term outcomes of neonatal insults.**

13.6–19.8). The frequency of impairment was highest for neurodevelopmental impairment [22.0% (*IQR* = 9.2–24.8)] and least for school problems [0.0% (*IQR* = 0.0–0.00)] in any of the conditions (Table 2).

In this meta-analysis, the highest prevalence of neurodevelopmental impairment was accounted for by HIE (31.6%: 95% CI 19.6–45.0) (Fig 2) and preterm birth (25.2%: 95% CI 15.1–36.8) (Fig 3). The prevalence of neurodevelopmental impairment in NNJ was 10.3% (95% CI 2.2–23.2) (Fig 4). Hearing impairment was most prevalent in congenital rubella (78.6: 95%CI 16.4–100.0) (Fig 5), while visual impairment was common in preterm birth (27.6%: 95% CI 20.6–35.2) (Fig 3). We conducted a subgroup analysis of 15 preterm birth studies with

**Table 1. Summary of Proportion of Impairment in Survivors of Neonatal Insults.**

| Neonatal insult | ParticipantsAssessed | Impaired n (%) | Type of Impairment n (%) | | | | | | | | | |
|---|---|---|---|---|---|---|---|---|---|---|---|---|
| | | | Neuro-develop-mental | Clinical | Mental | Hearing | Vision | Motor | Cerebral palsy | Behavior | Epilepsy | School problems |
| Bacterial Meningitis [27] | 111 | 41 (36.9%) | 41(36.9%) | - | - | - | - | - | - | - | - | - |
| Birth Asphyxia [28] | 54 | 13 (24.1%) | 13 (24.1%) | - | - | - | - | - | - | - | - | - |
| CMV [29–31] | 254 | 191 (75.2%) | 57 (22.4%) | - | - | 71 (27.9%) | 21 (8.3%) | 38 (15.0%) | 2 (0.8%) | - | 2 (0.8%) | - |
| Congenital Rubella [16, 17, 32–34] | 365 | 322 (88.2%) | 79 (21.6%) | 15 (4.1%) | 8 (2.2%) | 108 (29.6%) | 29 (7.9%) | | | 83 (22.7%) | - | - |
| Fetal Growth Restriction [35–38] | 2060 | 552 (26.8%) | 194 (9.4%) | - | 14 (0.7%) | - | - | - | 10 (0.5%) | 54 (2.6%) | - | 280 (13.6%) |
| HIE [39–46] | 500 | 292 (58.4%) | 128 (25.6%) | - | 70 (14.0%) | 4 (0.8%) | 3 (0.6%) | 26 (5.2%) | 23(4.6%) | 12 (2.4%) | 26 (5.5%) | - |
| Neonatal Jaundice [47–54] | 79,356 | 6586 (8.3%) | 6212 (7.8%) | - | - | 374 (0.5%) | - | - | - | - | - | - |
| Preterm birth [55–76] | 12,278 | 1531 (12.5%) | 1068 (8.8%) | 2(0.0) | 5(0.04) | 6 (0.1) | 108 (0.9%) | 133 (1.08%) | 69 (0.6%) | 117 (1.0%) | 5 (0.1%) | 3 (0.1%) |

CMV- congenital Cytomegalovirus infection

extremely low birth weight participants, and similar results were obtained. The least affected domains were mental health problems for fetal growth restriction (17.1%: 95% CI 12.6–22.2) (Fig 6) and neurological impairment in preterm birth (11.1%: 95% CI 6.1–17.2) (Fig 5). The meta-analysis was not possible for CMV due to the limited number of studies.

We did not find any statistically significant results from the meta-regression of the pooled proportion of impairment in preterm birth when the predictive factors of age at follow-up, region, and study design were included in the model. The funnel plot indicated an asymmetric plot confirming publication bias. The studies also had high heterogeneity which explains the asymmetry observed in the funnel plots.

## Discussion

This systematic review identified 52 studies that evaluated the long-term outcomes of NNI in school-aged children and older groups. Despite the limited data in these age groups, our study found 10% overall prevalence of impairment in survivors of NNI at age 6 years and above. The neurodevelopmental domain was the most commonly affected. Congenital rubella and HIE accounted for the highest frequency of impairment, while NNJ accounted for the least impairment.

Our findings that the neurodevelopmental domain is the most affected are similar to the results by Mwaniki and colleagues [8], however, the frequency was higher in their study (59% compared to 16.6% in the present study), and their samples comprised younger age-groups. These differences potentially demonstrate that neurodevelopmental impairment diminishes with age in most survivors due to the plasticity of the brain [8]. Another plausible reason may

**Table 2. Medians and Interquartile Range (IQR) of the Proportions of Impairment in Survivors of Neonatal Insults.**

| Type of Impairment | Overall | Neonatal Insults | | | | | | | |
|---|---|---|---|---|---|---|---|---|---|
| | | Bacterial Meningitis | Birth Asphyxia | CMV | Congenital Rubella | Fetal Growth Restriction | HIE | Neonatal Jaundice | Preterm birth |
| Neurodevelopmental | 22.0 (9.2–24.8) | 36.9 (36.9–36.9) | 24.1 (24.1–24.1) | 27.5 (8.2–46.9) | 32.7 (5.0–34.5) | 26.1 (15.6–31,7) | 34.1(19.8–48.1) | 11.6 (0.9–19.4) | 25.1 (14.1–42.0) |
| Clinical | 0.0 (0.0–0.0) | - | - | - | 30.0 (30.0–30.0) | - | - | - | 2.2 (2.2) |
| Mental | 0.0 (0.0–1.4) | - | - | - | 8.8 (7.5–10.0) | 11.4 (11.4–11.4) | 86.1 (86.1–86.1) | - | 5.6 (5.6–5.6) |
| Hearing | 0.3 (0.0–13.2) | - | - | 36.9 (15.0–58.7) | 72.1 (31.5–98.0) | - | 3.6 (3.6–3.6) | 0.5 (0.0–0.9) | 6.7 (6.7–6.7) |
| Vision | 0.3 (0.0–4.4) | - | - | 18.6 (18.6–18.6) | 29.8 (7.5–52.0) | - | 2.7 (2.7–2.7) | - | 24.6 (23.3–35.8) |
| Motor | 2.7 (0.0–3.1) | - | - | 33.6 (33.6–33.6) | - | - | 25.7 (15.3–36.0) | - | 10.0 (8.2–22.5) |
| Cerebral palsy | 0.8 (0.0–0.4) | - | - | 1.8 (1-8-1.8) | - | 1.9 (1.9–1.9) | 27.5 (23.1–32.0) | - | 5.5 (4.2–7.2) |
| Behavior | 0.5 (0.0–2.5) | - | - | - | 41.0 (33.7–48.3) | 10.1 (10.1–10.1) | 11.6(7.3–16.0) | - | 19.1 (18.1–21.0) |
| Epilepsy | 0.0 (0.0–0.4) | - | - | 1.8 (1.8–1.8) | - | - | 13.9 (13.5–14.3) | - | 5.6 (5.6–5.6) |
| School problems | 0.0 (0.0–0.0) | - | - | - | - | 14.2 (11.9–16.6) | - | - | - |

CMV- congenital Cytomegalovirus infection; HIE–Hypoxic Ischemic Encephalopathy; Entries are median (IQR).

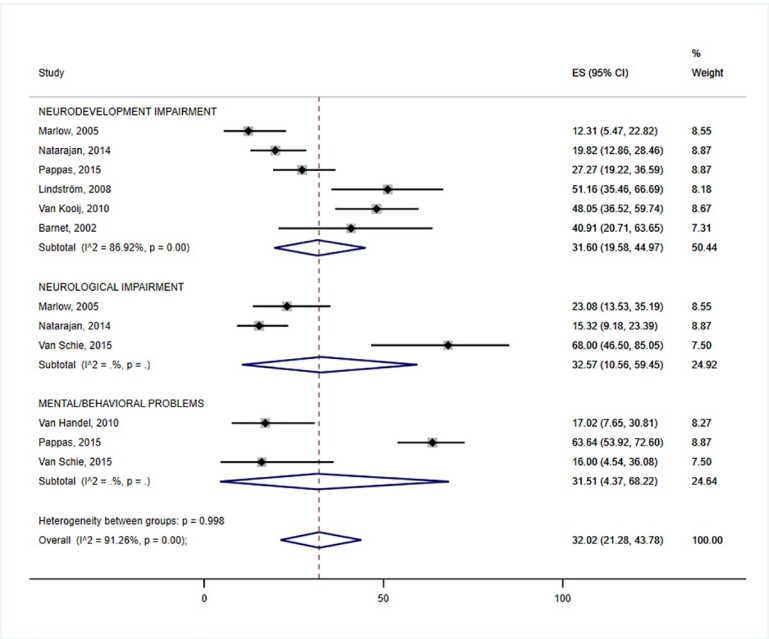

**Fig 2. Individual and pooled estimates and 95% confidence intervals for random-effects model examining the long-term outcomes of fetal growth restriction.** *I2*- heterogeneity statistic; ES- effect size; %—percent; sub-groups with (*I2* = . %, p = .) indicate that the number of studies were too few for the estimates to be calculated.

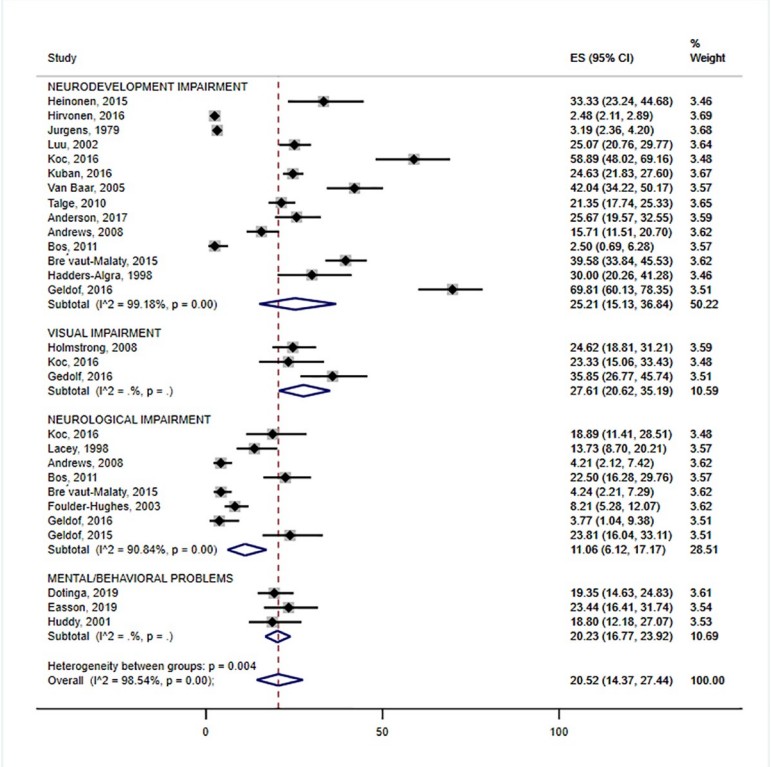

**Fig 3. Individual and pooled estimates and 95% confidence intervals for random-effects model examining the long-term outcomes of Preterm Birth.** *I2-* heterogeneity statistic; ES- effect size; %—percent; sub-groups with (*I2* = . %, p = .) indicate that the number of studies were limited for the estimates to be calculated.

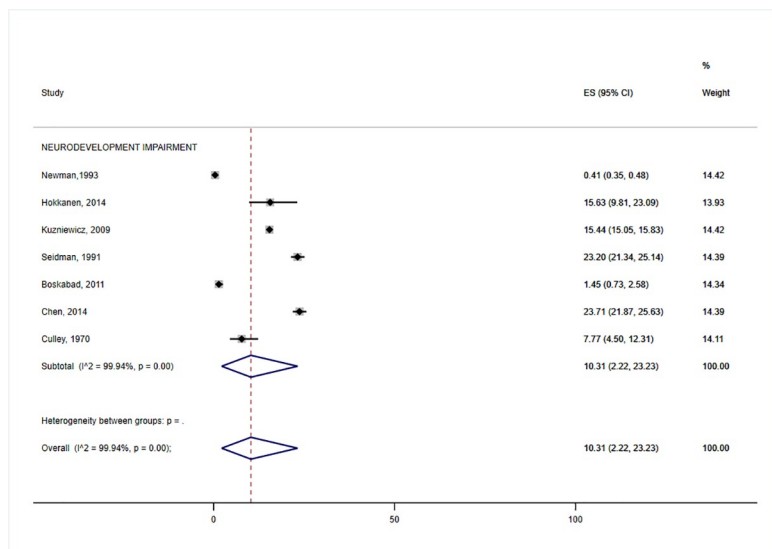

**Fig 4. Individual and pooled estimates and 95% confidence intervals for random-effects model examining the long-term outcomes of Neonatal Jaundice.**

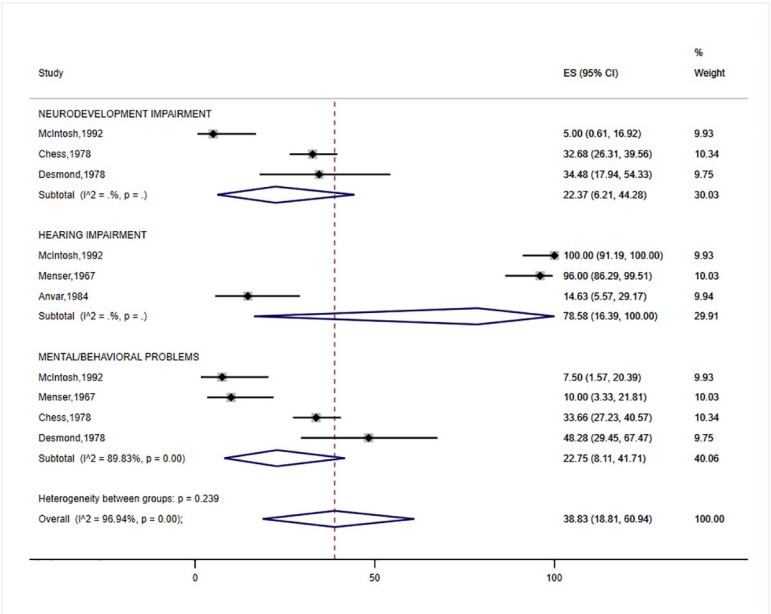

**Fig 5. Individual and pooled estimates and 95% confidence intervals for random-effects model examining the long-term outcomes of Congenital Rubella.** *I*2- heterogeneity statistic; ES- effect size; %—percent; sub-groups with (*I*2 = . %, p = .) indicate that the number of studies were limited for the estimates to be calculated.

be that children with severe neurodevelopmental impairment did not survive to older age, thereby leading to a diminished proportion observed in older ages. It is also likely that the burden of neurodevelopmental impairment in low and middle-income resource settings like SSA may not be well represented by existing research. Moreover, not all the studies in this current review have reported on all domains of impairment; it is possible that the burden in certain

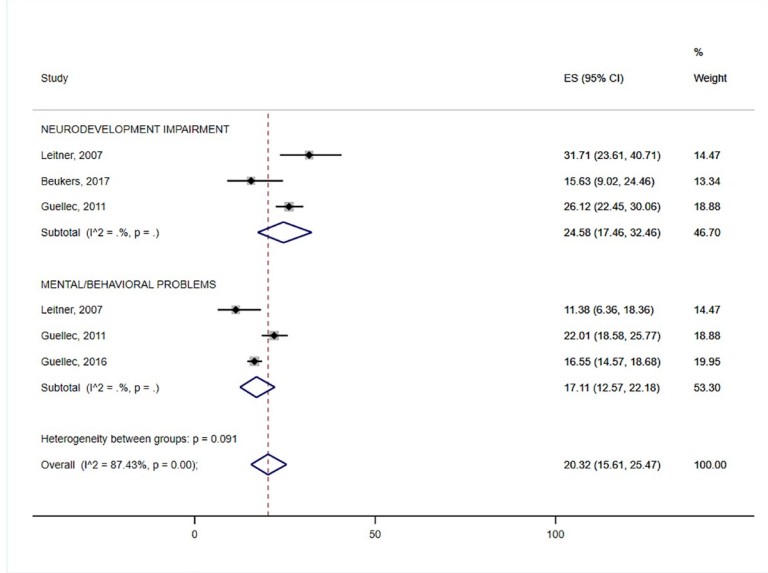

**Fig 6. Individual and pooled estimates and 95% confidence intervals for random-effects model examining the long-term outcomes of Hypoxic-ischemic encephalopathy.**

domains is underestimated. There is a need for more comprehensive longitudinal studies from SSA to better characterize the impact of NNI.

Similar to the results by Mwaniki and colleagues [8], congenital rubella and HIE were associated with the most frequent adverse long-term outcomes. A possible explanation regards to the pathogenesis of these conditions. Congenital rubella, affects the systemic organs of neonates [77] and causes cell and tissue damage, and is associated with ocular, heart, and hearing impairment. In individuals affected by HIE, the blood circulation or blood flow to the brain is impaired, leading to decreased oxygen and energy to the brain. This decrease in oxygen and energy can result in brain cell death and additional scar to the white matter or neuronal cell death in severe hypoxia [78], which may result in irreversible brain damage.

Although three-quarters of child deaths occur in SSA and South Asia, many of which are related to NNI, studies in Africa that examine the long-term outcome of NNI are scarce, and none was included in this review. The lack of research in these age groups could be attributed to the lack of appropriate assessment tools, inadequate access to health-care, or lack of research expertise in this area. This is unfortunate and has serious implications as school-aged children, or older age groups who survived NNI, but live with some degree of impairment may not receive the needed care and attention. Therefore, their potential outcomes are not fully optimized; hence, they may miss required support, such as getting specialized education and continue to be marginalized.

The major strengths of this study are the use of multiple databases and a focus on long-term outcomes that are often not well studied. Additionally, the study utilized a rigorous study design for systematic reviews and meta-analysis. Lastly, the study focuses on a global perspective and further sub-categorizes the domains of impairment, making it comprehensive.

However, this study has several limitations. First, the study inclusion criteria were strict and may have omitted some studies that could be included, e.g. studies that did not include a confirmed diagnosis in the first 28 days of life (e.g., vertical HIV infection). Second, the inclusion of all children with a diagnosis of NNJ or HIE in the study without considering the severity levels might have diluted our sample.

Third, in this review, a few studies reported multi-domain impairments in the patients. For instance, many participants with cerebral palsy will also have motor impairment and learning difficulties, which we found complex to conduct subgroup analysis. Fourth, the studies included were highly heterogeneous. However, we used the random effect model, which accounted for unexplained heterogeneity observed in the studies [79]. Lastly, the results of this study should be cautiously interpreted as we found publication bias indicated by the asymmetric forest plot, which may imply that studies with statistically significant results were more likely to be published than those with non-significant results. The publication bias could also reflect the high heterogeneity in the studies included in this systematic review and meta-analysis.

These results have several implications for research and prevention strategies. The review identifies that various forms of NNI have several adverse long-term outcomes, which necessitates further management and close monitoring of survivors of NNI. Despite the improvements in neonatal care and management of birth complications to prevent child mortality, there is still need for treatment and improved care for child survivors of NNI to protect the central nervous system and prevent adverse long-term outcomes. More resources should be allocated for longer follow-ups of survivors of NNI and preventive and rehabilitative services provided to them to optimize their outcomes. Long-term follow-up research is also needed to better understand the outcomes of NNI, especially in SSA where there is a high burden of NNI, and proper medical care is limited.

## Conclusion

NNI have long-term consequences that are experienced far beyond early childhood to older ages. The major impairments arising from NNI are recorded in the neurodevelopmental domains (cognition, language, developmental delay, and memory). However, this finding is based on studies from high-income countries such as Europe and America with the state of art of care for children. There are no robust studies on long-term outcomes of NNI in SSA despite the high burden of NNI in this region. Therefore, more investment of longer follow-ups of survivors of NNI is needed in SSA to optimize their outcomes fully.

## Supporting information

**S1 Checklist. PRISMA 2009 checklist.**
(DOC)

**S1 Appendix. Search terms.**
(DOCX)

**S2 Appendix. References of studies included in the review.**
(DOCX)

**S1 Table. Characteristics of studies included in the review.**
(DOCX)

**S2 Table. Assessment of quality of studies.**
(XLSX)

## Acknowledgments

We acknowledge permission from the Director of Kenya Medical Research Institute (KEMRI) to publish this work.

## Author Contributions

**Conceptualization:** Dorcas N. Magai, Carophine Nasambu, Derrick Ssewanyana, Charles R. Newton, Amina Abubakar.

**Data curation:** Dorcas N. Magai, Agnes M. Mutua, Esther Chongwo, Carophine Nasambu.

**Formal analysis:** Dorcas N. Magai, Eirini Karyotaki, Agnes M. Mutua, Esther Chongwo, Derrick Ssewanyana, Hans M. Koot.

**Investigation:** Dorcas N. Magai, Eirini Karyotaki, Agnes M. Mutua, Esther Chongwo, Derrick Ssewanyana, Charles R. Newton, Hans M. Koot, Amina Abubakar.

**Methodology:** Dorcas N. Magai, Eirini Karyotaki, Agnes M. Mutua, Esther Chongwo, Carophine Nasambu, Derrick Ssewanyana, Charles R. Newton, Hans M. Koot, Amina Abubakar.

**Project administration:** Dorcas N. Magai.

**Resources:** Dorcas N. Magai, Carophine Nasambu, Charles R. Newton, Amina Abubakar.

**Software:** Dorcas N. Magai, Esther Chongwo, Carophine Nasambu, Derrick Ssewanyana, Charles R. Newton, Hans M. Koot, Amina Abubakar.

**Supervision:** Eirini Karyotaki, Charles R. Newton, Hans M. Koot, Amina Abubakar.

**Validation:** Dorcas N. Magai, Eirini Karyotaki, Agnes M. Mutua, Carophine Nasambu, Derrick Ssewanyana, Charles R. Newton, Hans M. Koot, Amina Abubakar.

**Visualization:** Dorcas N. Magai, Agnes M. Mutua, Carophine Nasambu, Derrick Ssewanyana, Charles R. Newton, Hans M. Koot, Amina Abubakar.

**Writing – original draft:** Dorcas N. Magai.

**Writing – review & editing:** Dorcas N. Magai, Eirini Karyotaki, Agnes M. Mutua, Esther Chongwo, Carophine Nasambu, Derrick Ssewanyana, Charles R. Newton, Hans M. Koot, Amina Abubakar.

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
