## [Decision Letter · Decision Letter 0]

5 Feb 2020

PONE-D-19-22838

Long-term Outcomes of Survivors of Neonatal Insults: A Systematic Review and Meta-Analysis

PLOS ONE

Dear Dr Magai,

Thank you for submitting your manuscript to PLOS ONE. After careful consideration, we feel that it has merit but does not fully meet PLOS ONE’s publication criteria as it currently stands. Therefore, we invite you to submit a revised version of the manuscript that addresses the points raised during the review process.

See comments below 

We would appreciate receiving your revised manuscript by 4 March 2020. To enhance the reproducibility of your results, we recommend that if applicable you deposit your laboratory protocols in protocols.io, where a protocol can be assigned its own identifier (DOI) such that it can be cited independently in the future. For instructions see: http://journals.plos.org/plosone/s/submission-guidelines#loc-laboratory-protocols

We look forward to receiving your revised manuscript.

Kind regards,

Andrew Soundy

Academic Editor

PLOS ONE

Additional Editor Comments (if provided):

Thank you for your submission. Please provide a response to the reviewers comments. I have assessed the manuscript as a second reviewer and have no other comments to make. I note the quality of processes reported.

Journal Requirements:

2. Please ensure that you refer to Figure 4 in your text as, if accepted, production will need this reference to link the reader to the figure.

3. We note you have included a table to which you do not refer in the text of your manuscript. Please ensure that you refer to Table 2 in your text; if accepted, production will need this reference to link the reader to the Table.

Reviewers' comments:

Reviewer's Responses to Questions

**Comments to the Author**

1. Is the manuscript technically sound, and do the data support the conclusions?

Reviewer #1: Yes

2. Has the statistical analysis been performed appropriately and rigorously? 

Reviewer #1: I Don't Know

3. Have the authors made all data underlying the findings in their manuscript fully available?

Reviewer #1: Yes

4. Is the manuscript presented in an intelligible fashion and written in standard English?

Reviewer #1: Yes

5. Review Comments to the Author

Reviewer #1: Authors Magai et al present an interesting systematic review long-term outcomes of survivors of neonatal insults. The paper should be of interest to researchers and policy makers.

Introduction is concise and well written, aim is clear.

Methods:

Inclusion and exclusion criterias are clear, the search methods are appropriate. The review was registered in Prospero.

1) Is there a reason to choose the follow-up limit of 6 years?

Data extraction and analysis

Did you contact the authors for any missing data?

Results:

Line 120: 94,978 this number includes control group as well?

Line 123-124: please mention the % of survivors in bracket, example NNJ (83.5%)

Did you consider any NNJ or HIE as a neonatal insult? If yes your sample size is diluted. It would help if you include high risk insults such as NNJ requiring intensive phototherapy or exchange transfusion and moderate to severe HIEs.

Similarly a subgroup analysis of preterm birth by weight (ELBW, VLBW) will help.

Discussion

Overall discussion is well written, however following points may need attention.

1) Publication bias needs to be discussed

2) High heterogeneity is important. Random effect model may be useful, however a subgroup analysis of studies with lower heterogeneity is important (example I2 <60%)

3) No mention of any missing data and approach to address this important issue.

6. PLOS authors have the option to publish the peer review history of their article (what does this mean?). If published, this will include your full peer review and any attached files.

Reviewer #1: No

---

## [Author Response · Author response to Decision Letter 0]

6 Mar 2020

Review Comments to the Author

Methods:

1) Is there a reason to choose the follow-up limit of 6 years?

We considered children from six years of age, as this is the age where most children join elementary school, especially in low-and-middle-income countries. We have amended this part as follows:

Criteria for eligible studies were: i) empirical studies conducted in humans; ii) the study participants had follow-up data coinciding with the age bracket at six years or older (we considered children from six years as this is the age where most children join elementary school, especially in low-and-middle-income countries) (page 5, line 63-66).

Data extraction and analysis

2) Did you contact the authors for any missing data?

We had 65 articles which could not be accessed online. Most of them were older than two decades (probably not archived online). Out of these, in 50 articles we were not able to find the contact details of the corresponding authors, in one article our email was not delivered (the author had probably left the institution and changed their email addresses). Nine authors did not respond to our emails even after three reminders. Five authors sent us full articles; however, these did not meet the inclusion criteria.

Additionally, there were 35 articles with a wide age range (e.g. 2 months to 10 years). We contacted authors of these studies to inquire whether they conducted a sub-group analysis for participants aged six years and above. However, we were not able to get this information from the contacted authors. We have edited figure 1 to capture this information. 

Results:

3) Line 120: 94,978 this number includes control group as well?

The 94,978 were survivors of neonatal insults. The control group is not included in the numbers, as indicated below.

Overall, 94,978 survivors of NNI were included in the studies in this review (page 7, line 122).

4) Line 123-124: please mention the % of survivors in bracket, example NNJ (83.5%)

We have amended this part as follows: The most examined or studied NNI were HIE (15.4%), NNJ (15.4%), and preterm birth (42.3%) (page 8, line 126).

5) Did you consider any NNJ or HIE as a neonatal insult? If yes, your sample size is diluted. It would help if you include high risk insults such as NNJ requiring intensive phototherapy or exchange transfusion and moderate to severe HIEs.

The systematic review and meta-analysis aimed to examine the long-term impact of NNI globally. We considered all articles that provided a clear diagnosis of NNJ or HIE, and we did not limit the inclusion of these studies based on the severity of NNJ or HIE as we intended to determine if any of the children irrespective of the severity of their condition develop neurocognitive or mental health sequelae. Additionally, given the limited long-term outcome studies in these conditions, we included all the studies with a clear diagnosis of NNJ or HIE despite the severity of the condition. However, we have noted this as a limitation of our study as indicated below. 

Second, the inclusion of all children with a diagnosis of NNJ or HIE in the study without considering the severity levels might have diluted our sample (page 15, line 243-245).

6) Similarly, a subgroup analysis of preterm birth by weight (ELBW, VLBW) will help.

We have revised this part as follows: We conducted a sub group analysis of 15 preterm birth studies with extremely low birth weight participants, and similar results were obtained (page 8, line 138-139).

Discussion

7) Overall discussion is well written, however following points may need attention.

Publication bias needs to be discussed.

We have discussed the publication bias as follows:

Lastly, the results of this study should be cautiously interpreted as we found publication bias indicated by the asymmetry forest plot, which may imply that studies with significant results were more likely to be published than those with non-significant results. The publication bias could also reflect the high heterogeneity in the studies included in this systematic review and meta-analysis (page 15, line 251-255).

8) High heterogeneity is important. Random effect model may be useful, however a subgroup analysis of studies with lower heterogeneity is important (example I2 <60%)

We thank the reviewer for this suggestion and acknowledge his recommendation for the subgroup analysis. However, the studies included in the meta-analysis in each condition are few and not optimal to conduct a subgroup analysis. We appreciate the reviewer’s recognition of the robustness of the random effect model in dealing with heterogeneity in studies.

9) No mention of any missing data and approach to address this important issue.

In this systematic review and meta-analysis, we did not work with any primary data. We extracted secondary data from all the included studies, and we did not experience any missing data. Moreover, for the articles that we could not retrieve, their eligibility to be included in this study is doubtful since we are not sure whether they met the inclusion criteria for this systematic review and meta-analysis. We have corrected figure 1 to clarify that the 35 studies had a wide age range and not missing data as had previously been stated.

---

## [Editor Report · Decision Letter 1]

6 Apr 2020

Long-term Outcomes of Survivors of Neonatal Insults: A Systematic Review and Meta-Analysis

PONE-D-19-22838R1

Dear Dr. Magai,

We are pleased to inform you that your manuscript has been judged scientifically suitable for publication and will be formally accepted for publication once it complies with all outstanding technical requirements.

With kind regards,

Andrew Soundy

Academic Editor

PLOS ONE

Additional Editor Comments (optional):

Thank you for this resubmission.
---

## [Editor Report · Acceptance letter]

13 Apr 2020

PONE-D-19-22838R1 

Long-term Outcomes of Survivors of Neonatal Insults: A Systematic Review and Meta-Analysis 

Dear Dr. Magai:

I am pleased to inform you that your manuscript has been deemed suitable for publication in PLOS ONE. Congratulations! Your manuscript is now with our production department. 

With kind regards,

on behalf of

Dr. Andrew Soundy 

Academic Editor

PLOS ONE